# Single-step generation of homozygous knockout/knock-in individuals in an extremotolerant parthenogenetic tardigrade using DIPA-CRISPR

Koyuki Kondo [1,2], Akihiro Tanaka [1¤], Takekazu Kunieda [1]*

1 Department of Biological Sciences, Graduate School of Science, The University of Tokyo, Hongo, Bunkyo-ku, Tokyo, Japan, 2 Department of Life Science, Faculty of Advanced Engineering, Chiba Institute of Technology, Tsudanuma, Narashino, Chiba, Japan

¤ Current address: Department of Chromosome Science, National Institute of Genetics, Yata, Mishima, Shizuoka, Japan

* kunieda@bs.s.u-tokyo.ac.jp

**Data Availability Statement:** All data have been included in the manuscript and the supporting information.

## Abstract

Tardigrades are small aquatic invertebrates known for their remarkable tolerance to diverse extreme stresses. To elucidate the *in vivo* mechanisms underlying this extraordinary resilience, methods for genetically manipulating tardigrades have long been desired. Despite our prior success in somatic cell gene editing by microinjecting Cas9 ribonucleoproteins (RNPs) into the body cavity of tardigrades, the generation of gene-edited individuals remained elusive. In this study, employing an extremotolerant parthenogenetic tardigrade species, *Ramazzottius varieornatus*, we established conditions that led to the generation of gene-edited tardigrade individuals. Drawing inspiration from the direct parental CRISPR (DIPA-CRISPR) technique employed in several insects, we simply injected a concentrated Cas9 RNP solution into the body cavity of parental females shortly before their initial oviposition. This approach yielded gene-edited $G_0$ progeny. Notably, only a single allele was predominantly detected at the target locus for each $G_0$ individual, indicative of homozygous mutations. By co-injecting single-stranded oligodeoxynucleotides (ssODNs) with Cas9 RNPs, we achieved the generation of homozygously knocked-in $G_0$ progeny, and these edited alleles were inherited by $G_1/G_2$ progeny. This is the first example of heritable gene editing in the entire phylum of Tardigrada. This establishment of a straightforward method for generating homozygous knockout/knock-in individuals not only facilitates *in vivo* analyses of the molecular mechanisms underpinning extreme tolerance, but also opens up avenues for exploring various topics, including Evo-Devo, in tardigrades.

## Author summary

Tardigrades, tiny aquatic invertebrates also known as water bears, are celebrated for their extraordinary resilience to various extreme stresses like dehydration, radiation, and unusual

**Funding:** This work was supported by Japan Society for the Promotion of Science (JSPS; https://www.jsps.go.jp/english/) KAKENHI Grant Numbers JP20H04332, JP20K20580, JP21H05279 (to TK). AT received a Grant-in-Aid for JSPS Fellows (JP21J11385). The funder had no role in the study design, data collection and analysis, decision to publish, or preparation of the manuscript.

**Competing interests:** The authors have declared that no competing interests exist.

ranges of temperature and pressure. Understanding the molecular mechanisms of this resilience not only satisfies scientific curiosity but also holds promise for the development of innovative technologies for the dry preservation of biomaterials like biomedicines and vaccines. However, the lack of a heritable genome manipulation technology has hindered *in vivo* analyses of these mechanisms. This study addresses this longstanding challenge in the field. Employing an extremotolerant parthenogenetic tardigrade species, we established conditions that enable the efficient production of gene-manipulated individuals. Using these conditions, the simple injection of Cas9 genome-editing components into parental females leads to the generation of knockout/knock-in progeny. Unlike similar approaches in other animals, we obtained mutant progeny predominantly carrying a single type of mutation, namely, homozygous mutants, which significantly facilitates downstream analyses. This is the first report of a heritable gene-editing method in the entire group of tardigrades. The establishment of this straightforward method for generating gene-manipulated tardigrades not only facilitates *in vivo* analyses of the molecular mechanisms underpinning extreme tolerance, but also opens up avenues for exploring various topics, including Evo-Devo.

## Introduction

Tardigrades are microscopic invertebrates living in marine, limnic, and limno-terrestrial habitats. All of them require water in their surroundings to grow and reproduce. To date, more than 1,400 tardigrade species have been described [1]. Among them, some limno-terrestrial species are known to withstand the almost complete loss of water by entering a reversible ametabolic dehydrated state referred to as anhydrobiosis [2], and tardigrades can be stored in a desiccated state at room temperature, sometimes for over a decade [3]. Dehydrated tardigrades exhibit extraordinary resilience against various extreme stresses that would kill most other animals, such as low temperature (-273˚C), intense irradiation, the vacuum of space, and high hydrostatic pressure (7.5 GPa) [4–8]. This resilience is believed to arise from their remarkable cellular protection and repair mechanisms, which safeguard essential biomolecules and structures such as DNA, RNA, proteins, and membranes, supporting cellular functions. Understanding the molecular players and mechanisms involved in these processes not only satisfies scientific curiosity but also holds promise for the development of innovative technologies with significant implications for the storage and distribution of valuable but fragile biomaterials, like biomedicines and vaccines. Despite the growing interest in tardigrade resilience, the molecular mechanisms underlying this resilience have remained largely elusive. Some other desiccation-tolerant animals are known to accumulate and utilize non-reducing sugar, trehalose, as a vitrifying protectant against desiccation [9–11]. However, in anhydrobiotic tardigrades, trehalose accumulates at much lower levels or is even undetectable [12,13]. Instead, recently accumulating studies of tardigrades have suggested that they possess and utilize their own unique protective proteins whose expression is high and/or significantly induced upon desiccation during anhydrobiosis [14–17]. Owing to technological limitations, their functions and roles in tardigrade resilience have been elucidated largely using heterologous expression and/or *in vitro* systems [15,16,18–23]. Although RNAi is feasible for analyzing gene function and has been successfully used in some cases [18,24,25], the knockdown efficiency varied depending on the target gene and was not always sufficient. We developed a method of delivering Cas9 ribonucleoproteins (RNPs) to adult tardigrade cells in a largely transparent and anhydrobiotic tardigrade species, *Hypsibius exemplaris* [26]. By microinjecting Cas9 RNPs into the body cavity of adult tardigrades and subsequent electroporation, we demonstrated that gene

editing took place in some somatic cells of the injected tardigrades [26]. The same study also revealed that electroporation is not a prerequisite and the microinjection of Cas9 RNPs into the body cavity alone is sufficient to induce gene editing in some somatic cells in the tardigrades. However, tardigrade eggs are vulnerable to injection/needle-pricking [26], and the delivery to germline cells and subsequent generation of gene-edited individuals has not yet been achieved.

Recently, Shirai *et al.* (2022) developed a new gene-editing method termed direct parental CRISPR (DIPA-CRISPR) in cockroaches and red flour beetles [27]. Using DIPA-CRISPR, gene-edited progeny ($G_0$) can be obtained by simply injecting Cas9 RNPs into the hemocoel of parental female insects. The injected Cas9 RNPs are assumed to be incorporated into vitellogenic oocytes concomitantly with the massive uptake of yolk precursors. In agreement with this assumption, in DIPA-CRISPR it was shown to be critical for females to be injected at appropriate stages during vitellogenesis prior to the first oviposition. Our previous observations that injection alone was sufficient for the delivery of Cas9 RNPs to induce gene editing in somatic cells in the tardigrades and the successful generation of gene-edited progeny by DIPA-CRISPR in some insects prompted us to find out the appropriate conditions to enable the generation of gene-edited tardigrade individuals using a DIPA-CRISPR-like method.

In this study, we employed an anhydrobiotic and extremotolerant tardigrade, *Ramazzottius varieornatus* (Fig 1A–1C), because its genome sequence is available [16] and it lays eggs outside of exuviae, which helped us to collect eggs and obtain many individuals at the same age for injection. We particularly examined two critical parameters, the concentration of Cas9 RNPs and the age of females to be injected, both of which were quite different between our previous somatic cell gene editing in tardigrades and the original DIPA-CRISPR in insects [26,27]. By adjusting the conditions, we successfully obtained gene-edited progeny ($G_0$) for two target genes. *R. varieornatus* is a parthenogenetic species that lays eggs without mating. We found that most of the obtained gene-edited $G_0$ progeny carried the edited alleles in a homozygous form. In addition, we found that the simultaneous injection of single-stranded oligodeoxynucleotides (ssODNs) with the Cas9 RNPs led to the generation of knock-in progeny. To our surprise, the gene-editing efficiency in the knock-in trials was comparable to that in the knockout trials.

This study demonstrated that a DIPA-CRISPR-like method worked in an extremotolerant parthenogenetic tardigrade, *R. varieornatus*, and that the simple injection of Cas9 RNPs (+ knock-in donor if necessary) into parental tardigrades with the appropriate conditions is sufficient to obtain homozygous knockout/knock-in tardigrade individuals. This is the first example of heritable gene editing in the entire phylum of Tardigrada, and this gene-editing method should substantially promote *in vivo* analysis of the molecular mechanisms underpinning the extreme tolerance of tardigrades. In addition to their renowned resilience, tardigrades are becoming increasingly recognized as an emerging model for evolutionary and developmental biological study [28]. This is because the phylum Tardigrada has a close taxonomic relationship with two phyla containing super model invertebrates, *Drosophila melanogaster* (phylum Arthropoda) and *Caenorhabditis elegans* (phylum Nematoda), and would be expected to comprise suitable organisms for comparative study among them. Our method will also open up avenues for studying various Evo-Devo topics using tardigrades.

## Results

### Determination of Cas9 protein concentration for DIPA-CRISPR in *Ramazzottius varieornatus*

In DIPA-CRISPR, a relatively high concentration of Cas9 protein was used in the injection solution (3.3 μg/μL) compared with that in our previous tardigrade study (0.41 μg/μL; S1 Table) [26,27]; a lower concentration of Cas9 protein was reported to decrease the gene-

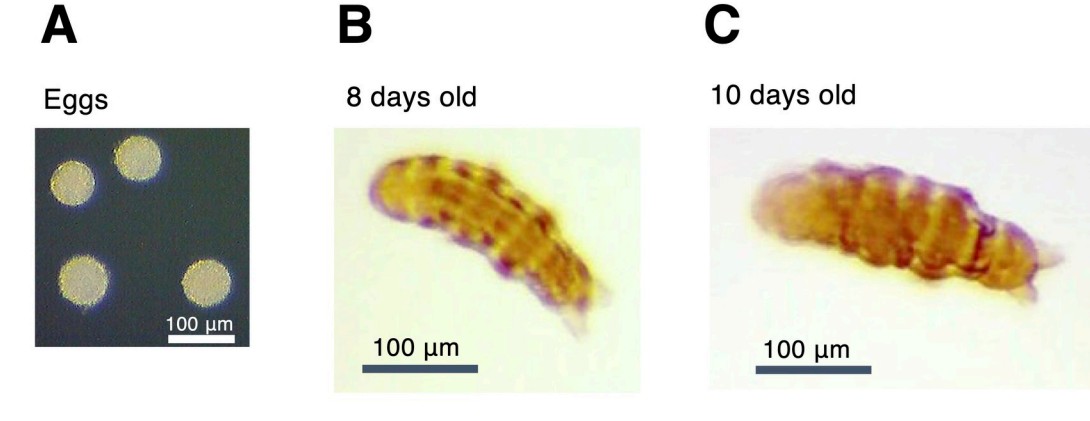

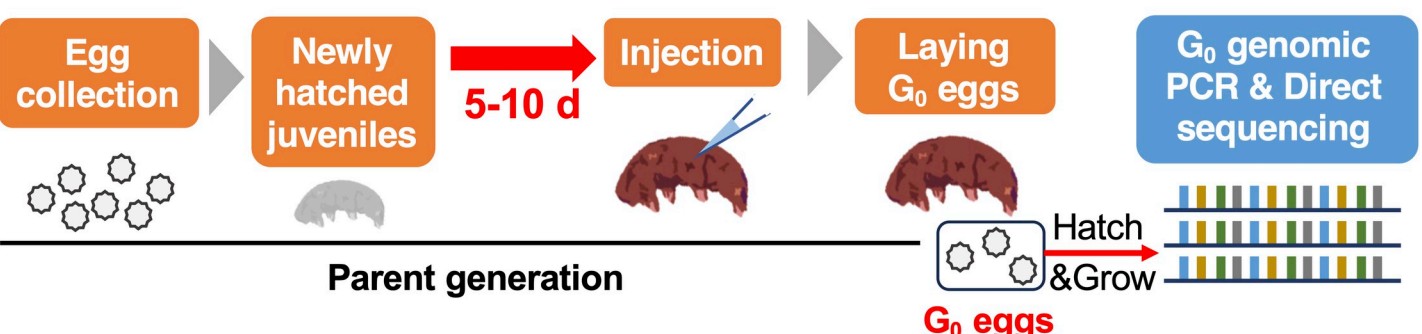

**Fig 1. Experimental scheme for DIPA-CRISPR in *R. varieornatus*.** Representative photographs of *R. varieornatus* eggs (A) and individuals at 8 (B) and 10 days old (C). (D) The stage of parental females to be injected is important for successful gene editing in DIPA-CRISPR. To obtain parental females at the defined age, eggs were collected and their hatching was examined daily. Newly hatched juveniles (0 days old) were separated and reared for the defined period prior to injection. After the injection of Cas9 RNPs, the injected tardigrades were reared for 10 days, and the laid eggs ($G_0$ progeny) were collected and reared. Grown $G_0$ individuals were separately subjected to genomic DNA extraction and PCR. PCR amplicons were directly analyzed by Sanger sequencing. Gene-edited $G_0$ progeny were obtained from parents injected at 7 to 10 days old.

editing efficiency [27]. Therefore, we attempted to increase the concentration of Cas9 protein in the injection solution. However, the commercial Cas9 protein solution usually contains a relatively high concentration of glycerol (e.g., 50% glycerol in IDT product), which could affect the viability of the injected animals. Accordingly, we first examined how a high concentration of glycerol can be tolerated by the injected tardigrades. As shown in S2 Table, the injection of 20% glycerol solution severely decreased the survival rate to 20%, while the survival rate remained at around half (45.5%) when using 15% glycerol solution. We thus chose to use a 15% glycerol concentration, which allows 3.0 μg/μL Cas9 protein in the injection solution, comparable to the level in the original DIPA-CRISPR method [27].

## Generation of gene knockout tardigrade individuals by DIPA-CRISPR

The experimental scheme of DIPA-CRISPR in tardigrades is shown in Fig 1D. In the original DIPA-CRISPR, the developmental stage of the parents to be injected was one of the most

critical parameters for successful gene editing in the progeny [27]. In most cases, the best stage is shortly before the first oviposition, which is consistent with the idea that Cas9 RNPs could be transported to oocytes concomitantly with the massive uptake of yolk precursors during vitellogenesis. Given that *R. varieornatus* usually starts to lay eggs around 10 days after hatching [8], we examined the period between 5 and 10 days after hatching for the injections into the tardigrades, as younger tardigrades (<5 days old) appeared to be too immature and were too small to be injected.

We chose the gene *RvY_01244* as a target, which encodes an ABC transporter belonging to the G subfamily (*ABCG*). Although some members of the ABCG family are known to be related to pigmentation in insects, such as *white*, *scarlet*, and *brown* genes in *Drosophila melanogaster* [29], phylogenetic analysis suggested that *RvY_01244* is not orthologous to those pigmentation-related members and its relationship to pigmentation was unclear (S1 Fig). To improve the gene-editing efficiency, we synthesized three crRNAs (Fig 2A) and injected RNP solution containing all three of them into parental tardigrades of each age from 5 to 10 days old. We expected that some intervening regions among the three crRNA targets would be deleted from the genome, which would be easily detected by examining the genomic PCR amplicon size. In total, we injected 414 parental tardigrades, 129 of which survived for more than 1 day (31.2% survival, Table 1). Using whole bodies of $G_0$ progeny, we successfully obtained genomic PCR amplicons for about 103 of 225 $G_0$ progeny and found one sample termed m1 that had a distinctly smaller amplicon size than that expected from the unmodified genome (Fig 2B). Direct sequencing of the short amplicon revealed complicated editing at the target locus. Specifically, the intervening region (205 bp) between crRNA1 and crRNA2 was lost and the 1,362 bp DNA fragment between crRNA2 and crRNA3 was re-inserted in the reverse orientation (Fig 2C). Notably, only the short amplicon was obtained from this sample (Fig 2B) and no mixed peaks were detected in the direct Sanger sequencing data. This suggested that this tardigrade carried the edited allele homozygously at the target locus, or carried another mutated allele that suppresses the PCR amplification around the target site (e.g., a huge deletion). Further direct sequencing of the remaining PCR amplicons with the same size as WT bands identified three additional gene-edited $G_0$ progeny, termed m2, m3, and m4. Among these mutants, m2 carried a 1-nt insertion at the crRNA1 cleavage site (Fig 2D) and m3 carried a 3-nt deletion at the crRNA3 cleavage site (Fig 2E). Again, almost no mixed peaks were detected in the direct Sanger sequencing data of both samples (Fig 2D and 2E), suggesting that both $G_0$ progeny were homozygous at the edited locus. We obtained similar results using a different primer set producing a longer amplicon, confirming the homozygosity of these edits (S2 Fig). The remaining m4 mutant carried two mutations. One was a 1-nt insertion at the crRNA1 site, which showed no mixed peaks, suggesting its homozygosity. The other one was a mutation at the crRNA3 site that exhibited partly mixed peaks in Sanger data, which could be interpreted as a mixture of two sequences: the unmodified genome and a 1-nt insertion at the cleavage site of crRNA3 (Fig 2F). The peak signals of the unmodified sequence were generally stronger than those of the 1-nt inserted sequence, suggesting that the 1-nt insertion might have occurred in a minor cell population during the development of this $G_0$ progeny, resulting in mosaicism.

Newly hatched juveniles of *R. varieornatus* are largely transparent, and their body gradually becomes brown as they grow. Among four obtained *RvY_01244* mutants, three of them, m1, m2, and m4, exhibited significant growth retardation. Two of them, m1 and m2 were sacrificed for genotyping at 10 and 12 days old, respectively, when they were very small and their body color was barely visible, while m4 was cultured for an extended period until 16 days old, at which point its body color became brown though its body was still much shorter than usual (S3A Fig). These three mutants carried frameshift mutations near the N-terminus of the target

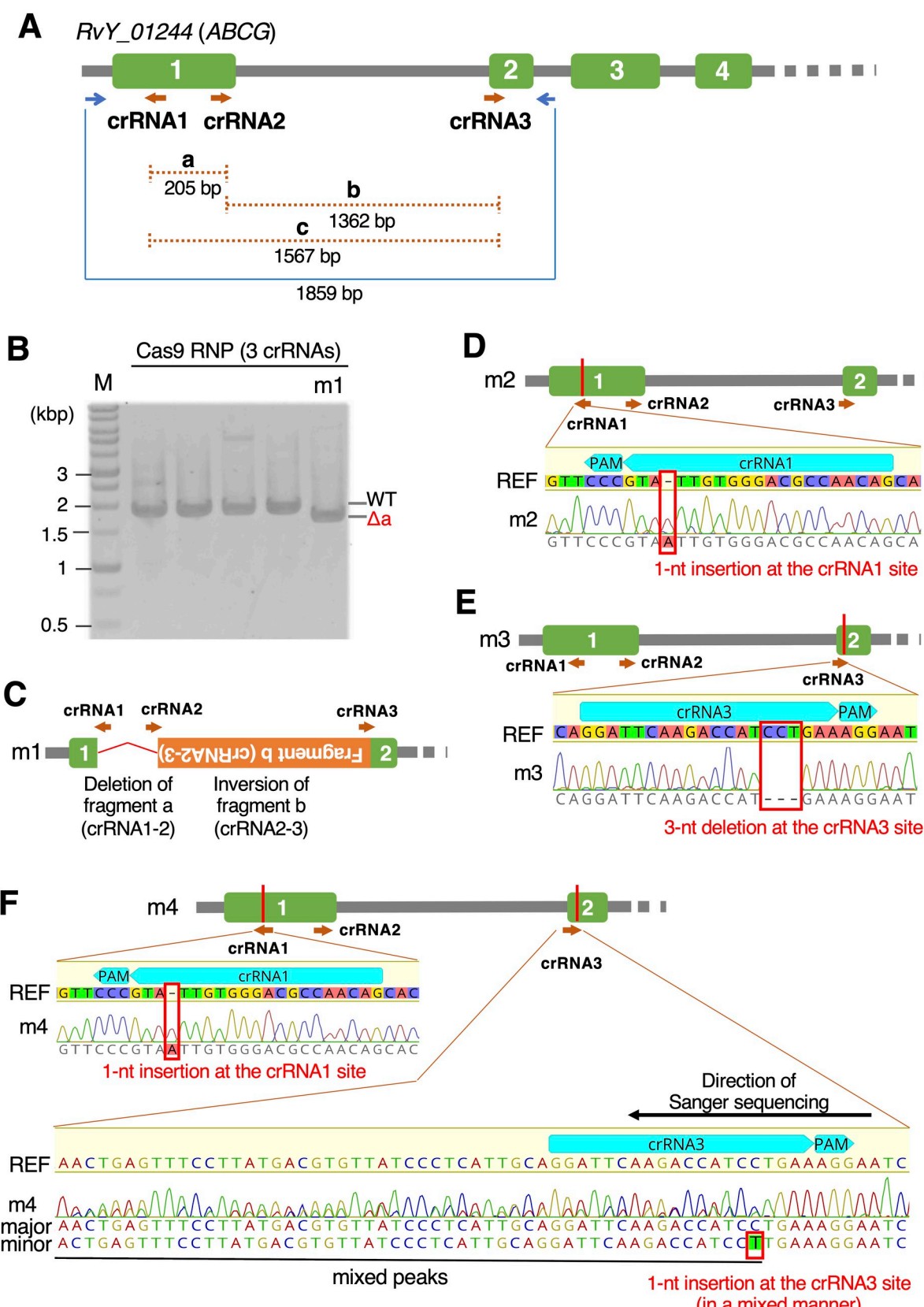

**Fig 2. Generation of G₀ progeny with gene editing at the *RvY_01244* gene locus.** (A) Schematic representation of the structure of the *RvY_01244* (*ABCG*) gene, and the locations of three crRNAs (brown arrows) and genomic PCR primers (blue arrows). Green boxes represent exons and gray lines represent introns or intergenic regions. (B) A representative agarose gel image of genomic PCR amplicons derived from some G₀ progeny. In this gel, the four samples on the left exhibited amplicons at the size expected from the unmodified genome (WT), while the right sample termed m1 exhibited a single band representing a shorter size (Δa) than WT, which roughly corresponds to the size with the deletion of fragment a (crRNA1–crRNA2). Note: The amplicon at the WT size was not detected in the m1 sample. (C–F) Gene-editing patterns in the four obtained gene-edited G₀ individuals, such as complex editing (m1, C), a 1-nt insertion (m2, D), a 3-nt deletion (m3, E), and two 1-nt insertions (m4, F). (C) Red bent line represents the deletion of the intervening region between crRNA1 and crRNA2. Orange box represents the intervening DNA fragment between crRNA2 and crRNA3, which was re-inserted in the reverse orientation. (D–F) Schematic representation of the gene-edited location and electropherograms in direct Sanger sequencing of genomic PCR amplicons with the reference sequence (REF).

protein and thus the protein function was likely disrupted. Meanwhile, no significant phenotype was observed in the m3 mutant carrying a 3-nt deletion, which causes the deletion of one amino acid without a frameshift. The observed phenotype appeared to be consistent with the severity of the corresponding mutations at the *RvY_01244* locus, but unexpectedly we frequently observed significant growth retardation even in many G₀ siblings carrying no edits at the target locus (S3B Fig). It is thus unclear whether the frameshift mutation at the *RvY_01244* gene causes growth retardation, but the observed phenotype of the m4 mutant suggested that brown coloration can proceed even with the frameshift in this gene.

Overall, the gene-editing efficiency (GEF; the proportion of gene-edited individuals among all sequenced individuals) identified here was 3.9% (Table 1). A total of four gene-edited G₀ progeny were obtained from the parental animals injected at 7 to 10 days old. Although this GEF is somewhat lower than in the original DIPA-CRISPR, our results indicated that DIPA-CRISPR works and can be used to generate gene knockout individuals in this tardigrade species.

## Editing of trehalose synthesis gene impaired hatchability of next generation

Next, we examined the general applicability of this methodology to other genes. As the next target, we chose *tps-tpp*, a gene responsible for trehalose synthesis. Trehalose is known to play important roles in desiccation tolerance in several anhydrobiotic animals, such as nematodes, a sleeping chironomid, and brine shrimps [9–11]. In tardigrades, however, trehalose

**Table 1. Summary of the gene editing targeting the *RvY_01244* (*ABCG*) gene.**

| Age (days) | # injected individuals | # survivors (ratio) | # G₀ eggs[a] | # genotyped G₀s | # gene edited G₀s (GEF) | Names of gene-edited G₀s |
|---|---|---|---|---|---|---|
| 5 | 29 | 9 (31.0%) | 6 | 5 | | |
| 6 | 43 | 11 (25.6%) | 20 | 9 | | |
| 7 | 60 | 17 (28.3%) | 12 | 9 | | |
| 7–8 | 13 | 6 (46.2%) | 7 | 4 | | |
| 8 | 112 | 26 (23.2%%) | 34 | 24 | 2 (8.3%) | m1, m2 |
| 7–9 | 39 | 2 (5.1%) | 6 | 3 | 1 (33.3%) | m4 |
| 8–9 | 10 | 5 (50%) | 14 | 2 | | |
| 9 | 30 | 12 (40%) | 27 | 5 | | |
| 9–10 | 49 | 27 (55.1%) | 63 | 30 | 1 (3.3%) | m3 |
| 10 | 29 | 14 (48.3%) | 36 | 12 | | |
| Total | 414 | 129 (31.2%) | 225 | 103 | 4 (3.9%) | |

[a] Laid by the injected animals within 10 days after injection.

production is not a common feature in anhydrobiotic species and the trehalose synthesis gene has been found in only two lineages: superfamily Macrobiotoidea and genus *Ramazzottius* [30]. A previous comprehensive phylogenetic analysis of the *tps-tpp* gene in the animal kingdom suggested that two tardigrade lineages, one of which includes *R. varieornatus*, have independently acquired distinct bacterial trehalose synthesis genes via horizontal gene transfer [30]. *R. varieornatus* has a single *tps-tpp* gene (*RvY_13060*), which encodes a fusion enzyme of trehalose-6-phosphate synthase (TPS) and trehalose-6-phosphate phosphatase (TPP). This enzyme is sufficient to produce trehalose from glucose-6-phosphate and UDP-glucose [16,30]. We designed two crRNAs targeting exon 8 or exon 9 of the *tps-tpp* gene, both of which are located within the TPS domain (Fig 3A), and injected RNP solution containing both of them into parental tardigrades from 7 to 10 days old. As shown in Table 2, we obtained five $G_0$ progeny carrying edited genes. Of these, one was the offspring of a parent injected at 7 days old and four were offspring of those injected at 10 days old. In total, GEF was 3.4%. In all examined $G_0$ progeny including the gene-edited ones, genomic PCR amplicons were essentially detected as a single band in agarose gel electrophoresis (Fig 3B). Sanger sequencing of the amplicons revealed that four of the five gene-edited $G_0$ progeny carried distinct insertions or deletions without apparent mixed peaks, suggesting that they carried homozygous mutations (Figs 3C and S4A–S4D). The remaining one exhibited partly mixed peaks in Sanger data, which could be interpreted as a mixture of two sequences, namely, the unmodified genome and a 1-nt deletion at the cleavage site of crRNA1, although both sequences commonly carried a 333 bp deletion at the cleavage site of crRNA2 (Fig 3C and 3D). The peak signals of the unmodified sequence were generally more intense than those of the sequence with a 1-nt deletion, suggesting that the 1-nt deletion might have occurred in a minor cell population during the development of this $G_0$ progeny, resulting in mosaicism, while the 333 bp deletion likely occurred in the oocyte stage, resulting in its homozygosity.

Genotyping data shown in Fig 3B–3D indicated successful generation of $G_0$ progeny carrying the *tps-tpp*-knockout mutations by DIPA-CRISPR. To examine the effects of *tps-tpp* knockout on tardigrade physiology, we next attempted to establish *tps-tpp*-knockout strains by rearing $G_0$ individuals until they laid $G_1$ eggs before sacrifice for genotyping. We again injected Cas9 RNPs with the two same crRNAs targeting *tps-tpp* into parental tardigrades aged 7 to 10 days old. After rearing the $G_0$ progeny until they laid $G_1$ eggs, we analyzed the genome sequence of each $G_0$ progeny. As shown in Table 3, we obtained six $G_0$ progeny carrying the edited genes among 151 examined individuals (GEF = 4.0%). Of those, four individuals had the same editing, which was a 1-nt insertion at the crRNA2 cleavage site (Fig 3E). One of the other two remaining individuals had 8-nt and 3-nt deletions at the crRNA1 and crRNA2 cleavage sites, respectively (Fig 3E), while the other one had a 484-nt deletion between crRNA1 and crRNA2 (Fig 3E). Again, in all edited $G_0$ individuals, only a single amplicon was detected in agarose gel electrophoresis, and no mixed peaks were detected in direct Sanger sequencing (S4E and S4F Fig). We also re-performed PCR using a different primer set from some genomic DNA samples of the gene-edited $G_0$ progeny and obtained the same results: no mixed peaks were present in those amplicons (S5 Fig). Each $G_0$ individual carrying the edited genes laid several $G_1$ eggs (2–8 eggs/$G_0$ individual), with 24 eggs in total from six $G_0$ individuals (S3 Table). However, unexpectedly, all of the $G_1$ eggs from the gene-edited $G_0$ individuals failed to hatch (hatching rate: 0%; S3 Table). Meanwhile, the hatchability of $G_1$ eggs laid by $G_0$ individuals with no editing was 89.5% (S4 Table). These observations suggested that the editing of the *tps-tpp* gene impaired the hatchability of the $G_1$ progeny in *R. varieornatus*.

## Establishment of gene knock-in strains by DIPA-CRISPR

The CRISPR-Cas9 system including DIPA-CRISPR has been used to generate not only gene knockout individuals, but also knock-in ones, which enables precise modification of the target

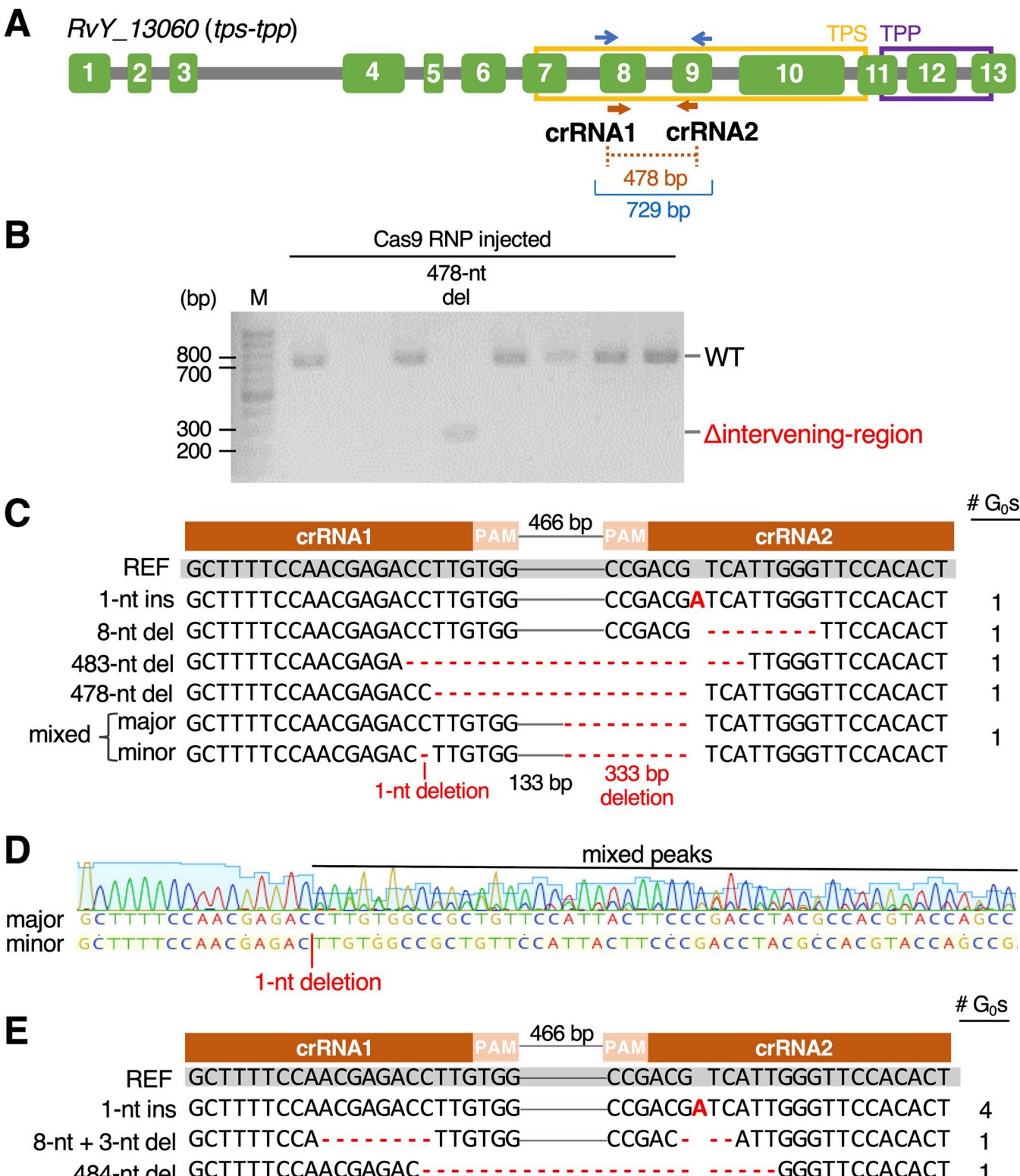

**Fig 3. Generation of *tps-tpp* knockout tardigrades.** (A) Schematic representation of the structure of the *RvY_13060* (*tps-tpp*) gene, and the locations of two crRNAs (brown arrows) and genomic PCR primers (blue arrows). Green boxes represent exons and gray lines represent introns. (B) A representative agarose gel image of genomic PCR amplicons from some $G_0$ progeny. 'WT' indicates the amplicon size predicted from the unmodified genome and 'Δintervening-region' indicates the size with the deletion of the intervening region between two crRNAs. The sample labeled '478-nt del' exhibited a single amplicon at a size corresponding to a 478-nt deletion. (C) Comparison of the amplicon sequences in five gene-edited $G_0$ progeny with the reference sequence (WT). The

numbers of $G_0$ individuals carrying each editing pattern are shown in the right column. Bold red letters and hyphens indicate insertions and deletions. In Sanger sequencing data, four gene-edited $G_0$ individuals clearly exhibited a single sequence without mixed peaks (S4A–S4D Fig). The other one exhibited mixed sequences of the unmodified one (major) and a 1-nt deletion (minor) at the crRNA1 cleavage site, while both of the mixed sequences shared the same 333 bp deletion around the crRNA2 cleavage site. (D) Electropherograms of direct Sanger sequencing of the gene-edited $G_0$ individual containing mixed peaks. The Sanger data were obtained using the forward primer (left in panel A). There are no mixed peaks in the left portion prior to the putative 1-nt deletion site. In contrast, in the right portion, minor peaks derived from the 1-nt deletion sequence were detected with the major peaks corresponding to the unmodified sequence. (E) Gene-editing patterns in the gene-edited $G_0$ progeny whose $G_1$ eggs were successfully obtained for further analyses. The numbers of $G_0$ individuals carrying each editing pattern are shown on the right. Each $G_0$ individual exhibited only one kind of edited sequence (S4E and S4F Fig), indicative of homozygous mutation. The intervening 466 bp regions between crRNA1 and crRNA2 are shown by thin gray lines.

genome region as designed. To investigate whether the method above is applicable for gene knock-in in tardigrades, we co-injected single-stranded oligodeoxynucleotides (ssODNs) with Cas9 RNPs into tardigrades aged 7 to 10 days old. We again targeted the gene $RvY\_01244$ ($ABCG$). We designed the crRNA near the C-terminus of the coding sequence and the ssODNs to introduce 11 separate single-nucleotide substitutions; 10 of them were synonymous mutations, including two mutations in PAM, while the other one changed the amino acid from valine (GTG) to methionine (ATG) (Fig 4A). As shown in Table 4, we obtained five $G_0$ progeny carrying edited genes out of 107 examined $G_0$ individuals (GEF = 4.7%). Three of them exhibited a clear single sequence without mixed peaks in Sanger sequencing, in which every nucleotide at the 11 positions was completely substituted as designed in the ssODNs (Fig 4B). This suggested that they carried the knocked-in allele in a homozygous manner. Sanger sequencing data of another gene-edited individual exhibited a mixture of the fully knocked-in sequence as a major peak and the unmodified (WT) sequence as a minor one (Fig 4B), suggesting the mosaicism of the individual. The remaining individual exhibited a more complicated pattern; it carried a 1-nt insertion at the crRNA cleavage site in a homozygous manner, and also exhibited mixed peaks of the knocked-in sequence and the unmodified sequence at the two modification sites furthest from the crRNA cleavage site (Fig 4B).

To examine whether these edited alleles are heritable, we examined the genotypes of $G_1/G_2$ progeny of these gene-edited $G_0$ individuals after propagation. $G_1$ progeny were separately reared and subjected to genotyping after laying $G_2$ eggs. From a perfectly knocked-in $G_0$ individual, one $G_1$ progeny was successfully reared and propagated, in which all of the examined

**Table 2. Summary of the gene editing targeting the $RvY\_13060$ ($tps$-$tpp$) gene.**

| Age (days) | # injected individuals | # survivors (ratio) | # $G_0$ eggs[a] | # genotyped $G_0$s | # gene edited $G_0$s (GEF) |
|---|---|---|---|---|---|
| 7 | 93 | 32 (34.4%) | 44 | 36 | 1 (2.8%) |
| 8 | 130 | 31 (23.8%) | 54 | 43 | |
| 9 | 77 | 14 (18.2%) | 17 | 15 | |
| 10 | 41 | 15 (36.6%) | 73 | 52 | 4 (7.7%) |
| Total | 341 | 92 (27.0%) | 188 | 146 | 5 (3.4%) |

[a] Laid by the injected animals within 10 days after injection.

**Table 3. Summary of generation of $tps$-$tpp$ gene-edited $G_0$ individuals laying $G_1$ progeny.**

| Age (days) | # $G_0$ eggs[a] | # hatched $G_0$ eggs (ratio) | # genotyped $G_0$s which laid $G_1$ eggs | # gene edited $G_0$s (GEF) |
|---|---|---|---|---|
| 7–10 | 182 | 165 (90.7%) | 151 | 6 (4.0%) |

[a] Laid by injected animals within 10 days after injection.

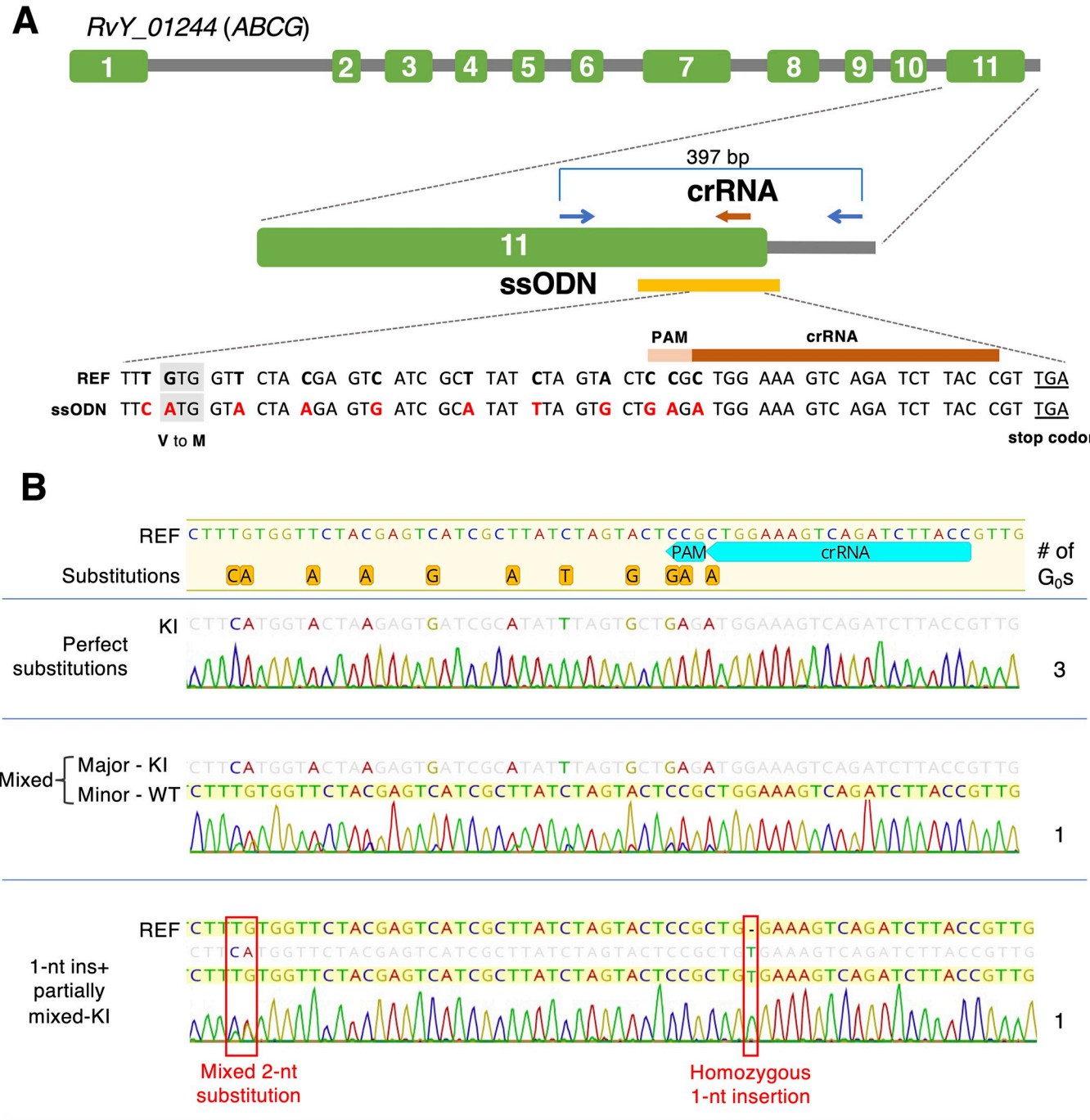

**Fig 4. Generation of gene knock-in tardigrades.** (A) Schematic representation of the structure of the *RvY_01244* (*ABCG*) gene and the locations of crRNA (brown arrows), genomic PCR primers (blue arrows), and ssODNs (yellow line). Green boxes represent exons and gray lines represent introns and intergenic regions. The ssODN sequence into which the 11 substitutions (red letters) were introduced is shown in alignment with the reference sequence (REF). (B) Gene-editing patterns in the gene-edited $G_0$ progeny obtained by co-injecting ssODNs and their representative electropherograms in direct Sanger sequencing of the amplicons. The number of $G_0$ individuals with each editing pattern is shown on the right. The three $G_0$ individuals exhibited a clear single sequence carrying the 11 separate single-nucleotide substitutions as designed in ssODNs (perfect substitutions). One of the other $G_0$ individuals (shown in the middle of panel B) exhibited a mixture of the sequence with 11 separate single-nucleotide substitutions (knocked-in, KI) as a major peak and the unmodified sequence (WT) as a minor one. The other one (shown in the bottom of panel B) carried a 1-nt insertion at the cleavage site and also exhibited two consecutive mixed peaks of unmodified and knocked-in sequences at the knock-in position furthest from the cleavage site.

**Table 4. Summary of knock-in experiments targeting *RvY_01244* (*ABCG* gene).**

| Age (days) | # injected individuals | # survivors (ratio) | # G$_0$ eggs[a] | # hatched eggs (ratio) | # genotyped G$_0$s | # gene edited G$_0$ (GEF) |
|---|---|---|---|---|---|---|
| 7–10 | 210 | 73 (34.8%) | 137 | 125 (91.2%) | 107 | 5 (4.7%) |

[a] Laid by injected animals within 10 days after injection.

G$_1$ and G$_2$ progeny were confirmed to carry the perfectly knocked-in allele in the homozygous form (S6 Fig and S5 Table). In terms of their appearance, those homozygous knock-in individuals looked similar to those carrying no edits (S7 Fig). From the G$_0$ individual carrying a mixture of the knocked-in sequence as a major peak and the unmodified sequence as a minor one, five G$_1$ progeny were obtained. Although one G$_1$ progeny could not be genotyped owing to amplification failure, the other four G$_1$ progeny were confirmed to carry the fully knocked-in sequence homozygously and G$_2$ progeny of each G$_1$ individual were confirmed to carry the same knock-in sequence (S5 Table). Thus, the observed knocked-in sequence was successfully inherited by the progeny. From the remaining G$_0$ progeny that carried the homozygous 1-nt insertion and mosaic 2-nt knocked-in sequence, one G$_1$ egg that carried only the 1-nt insertion was obtained. The detected 2-nt knock-in sequence as a mosaic did not appear to be heritable, while the homozygous 1-nt insertion was heritable (S5 Table).

## Discussion

In this study, we demonstrated that DIPA-CRISPR successfully worked in an extremotolerant parthenogenetic tardigrade. In the original DIPA-CRISPR, the injected Cas9 RNPs are assumed to be incorporated into vitellogenic oocytes concomitantly with the massive uptake of vitellogenins by receptor-mediated endocytosis. Thus, injecting the individuals at appropriate developmental stages was one of the critical parameters for successfully obtaining gene-edited progeny [27]. As we have no knowledge of the vitellogenic process in *R. varieornatus*, we injected parental tardigrades at 5 to 10 days old, which corresponds to the stage just before the first oviposition, and obtained gene-edited progeny from the parents injected at 7 to 10 days old. In a related tardigrade species, *H. exemplaris*, which belongs to the same taxonomic family as *R. varieornatus*, the vitellogenic process appears to consist of three distinct modes lasting 4 days: the first part of the yolk is synthesized by the oocyte itself (autosynthesis); the second part is synthesized by trophocytes and transported to the oocyte through cytoplasmic bridges; and the third part is synthesized outside the ovary and transported to the oocyte by endocytosis [31]. In this three-step process of vitellogenesis, the injected Cas9 RNPs could be incorporated into oocytes during the third stage. In *R. varieornatus*, the germ cells could take up the injected Cas9 RNPs in this way.

*R. varieornatus* is a diploid parthenogenetic species [8,16], but its cytological processes of progeny production and the mode of inheritance of genetic materials have remained unclear. Ammermann (1967) reported the cytological processes of diploid parthenogenetic reproduction in a related tardigrade species, *Hypsibius dujardini*, which is a species complex containing the recently redescribed *H. exemplaris* and belongs to the same taxonomic family as *R. varieornatus* [32,33]. During oogenesis in *H. dujardini*, the female germ cells undergo the first meiosis and daughter cells receive the mostly homozygous dyads derived from the meiotic bivalent chromosomes. After that, the dyad disintegrates and the diploidy is recovered in the daughter cells. Meiosis is completed by the subsequent mitosis-like process of the second meiosis, which maintains the diploidy. According to this cytological process, the chromosomes of the oocytes are predicted to be largely homozygous, except the small possible heterozygous regions that could be derived from chromosomal crossover during the first meiosis. In our genotyping

analyses of the edited $G_0$ individuals of *R. varieornatus*, only a single sequence was detected in most direct Sanger sequencing, suggesting that most $G_0$ progeny carried the edited allele in a homozygous manner (Figs 2B–2E, 3B, 3C, 3E, 4B, S2, S4 and S5). Notably, a similar result was obtained in the case of the very complicated editing of the *RvY_01244* (*ABCG*) gene by three crRNAs, in which one intervening region was deleted and the other intervening fragment was re-inserted in the reverse orientation (Fig 2C). It is unlikely that the Cas9 RNPs independently performed the same complicated editing on both alleles in a germ cell. Thus, this result is very difficult to explain if *R. varieornatus* undergoes clonal (ameiotic) propagation. If the cytological process of parthenogenetic reproduction in *R. varieornatus* is similar to that in *H. dujardini*, it is assumed that the CRISPR-Cas9 system would edit the single allele in a germ cell before meiosis, and the mutation would then be replicated and transferred to the mature egg cell in a homozygous form during the meiotic process (S8 Fig). This is good news for researchers because a homozygous mutant could be obtained in a single step and can be parthenogenetically propagated without the need for further crossing, which significantly facilitates downstream analyses. In some haplodiploid arthropods, males carry a haploid genome, although females are diploid, and thus the similar application of CRISPR system to parental females often produces mutant male progeny carrying a single mutant allele hemizygously [34,35]. However, in these cases, multiple crossings and selections are needed to establish homozygous mutant strains. In parthenogenetic species, it is generally difficult to apply Mendelian genetic approaches, but some species like *R. varieornatus* might have an advantage in reverse genetics. In our genotyping data, a few cases showed weak mixed peaks (Figs 2F, 3C, 3D and 4B). We assumed that these minor peaks were derived from mosaic mutations, which might occur via the delayed action of the remaining Cas9 RNPs in a small cell population during the development of $G_0$ progeny.

In general, gene knock-in mediated by homology-directed repair (HDR) tends to occur at a much lower rate than gene knockout mediated by nonhomologous end-joining (NHEJ) [36–39], although germ cells are more prone to HDR than somatic cells [40]. In the original DIPA-CRISPR, the proportion of edited individuals among the total number of individuals hatched in knock-in trials was 1.2%, while it was 50.8%–71.4% in knockout trials in red flour beetles at the optimized stages [27]. In this study, however, knock-in efficiency was comparable to that of knockout (4.7% and 3.4%–4.0%, respectively) and we rarely observed short indels by NHEJ-mediated repair in the knock-in experiments. This suggested that HDR might be a dominant repair mode in the germ cells of this tardigrade species. Notably, using *R. varieornatus* (in this study), we did not observe the tendency for no-indel NHEJ that was observed in somatic cells of *H. exemplaris* in our previous study [26]. This could be consistent with the relatively low efficiency of NHEJ-dependent repair (knockout) in this study.

In gene-editing experiments targeting the *RvY_01244* (*ABCG*) gene, no significant phenotype was observed in the mutants carrying single amino acid deletion near the N-terminus (m3) or substitution near the C-terminus (knock-in), suggesting that these mutations did not significantly affect the gene function of *RvY_01244*. The mutants carrying frameshift mutations near the N-terminus (m1, m2, and m4) exhibited growth retardation and slow coloration compared with uninjected individuals, but a similar phenotype was observed in the $G_0$ siblings carrying no edits at the target locus (S3 Fig). It remains unclear why the unedited siblings exhibited similar growth retardation. We cannot rule out the possibility that these phenotypes could be derived from an off-target effect, although all designed crRNAs (even 12-mer near PAM) exhibited a unique match in the genome sequence. As shown in S3 Fig, brown coloration could proceed even in the m4 mutant carrying a frameshift mutation in *RvY_01244*. These results could be consistent with the phylogenetic analysis in which RvY_01244 was shown to be non-orthologous to pigment-related ABCG members like *white*, *brown*, and *scarlet* (S1 Fig).

In all *tps-tpp*-edited mutants obtained in this study, a frameshift was introduced at the putative cleavage sites of crRNA1 or crRNA2, both of which were located within the TPS domain (Fig 3A). Thus, the mutated *tps-tpp* gene products likely lost the function of the C-terminal region of TPS and the whole of TPP (Fig 3A). Because the C-terminal region of TPS is responsible for the binding to the substrate UDP-glucose [41], the TPS activity was likely lost in the edited tardigrades as well. In all *tps-tpp*-edited mutants, $G_0$ individuals were able to hatch, grow, and lay eggs normally, but no $G_1$ eggs hatched (S3 Table). The hatchability of $G_1$ eggs was significantly lower in *tps-tpp*-edited mutants than in those harboring no editing in the *tps-tpp* gene ($p$ = 2.57e-15, Fisher's exact test). These results suggested that the mutations in the *tps-tpp* gene had a maternal effect on the hatchability of the embryos in this tardigrade species. For instance, trehalose could be synthesized in maternal tissue and transported to the oocyte and might play important roles in embryogenesis of the progeny, for example, as an energy reserve. In the cockroach *Periplaneta americana*, treatment with the trehalase inhibitor validoxylamine A (VAA) inhibited normal oocyte development, indicating that trehalose is necessary for successful oocyte development in this insect species [42]. However, we would like to keep the question open of whether trehalose itself plays an important role in tardigrade physiology. This is because the TPS-TPP protein of *R. varieornatus* contains an extraordinarily long N-terminal region, which exhibits sequence similarity with trans-1,2-dihydrobenzene-1,2-diol dehydrogenase. This N-terminal region could be translated even in the *tps-tpp* mutants and the truncated gene products might be harmful and responsible for the observed phenotype in this study, instead of trehalose reduction.

In summary, we have successfully established a method for generating both gene knockout and gene knock-in individuals in the anhydrobiotic and extremotolerant tardigrade species *R. varieornatus*, by adjusting the conditions of DIPA-CRISPR. Our findings indicated that the optimal injection window is between 7 and 10 days after hatching, aligning with the period shortly before the first oviposition in this species. The simple injection of Cas9 RNPs (with knock-in donor when necessary) into parental tardigrades at the appropriate age is sufficient to obtain the edited progeny. Notably, such progeny predominantly carried the edited allele in homozygous form, which was probably attributable to the meiotic parthenogenetic mode of reproduction. This feature significantly facilitates loss-of-function analyses downstream. While DIPA-CRISPR and similar methods were initially developed in arthropods [27,43], our study shows its effectiveness in a non-arthropod organism, underscoring the broad applicability of this method to various invertebrate species, including other tardigrades. This method should facilitate *in vivo* analysis of various topics in tardigrades, including the molecular mechanisms underlying their renowned extreme tolerance as well as many Evo-Devo-related issues.

## Materials and methods

### Tardigrades

*Ramazzottius varieornatus* YOKOZUNA-1 strain was reared as described previously [22]. Briefly, tardigrades were maintained in 1.2% agar plates overlaid with sterilized pure water (Elix Advantage 3 UV, Millipore) containing live chlorella suspension (Recenttec) as food at 22˚C. Water and food were replaced once a week. To prepare the staged tardigrades for injection, eggs were collected from culture dishes and transferred to a new agar dish with food after cleaning treatment with 1% commercial chlorine bleach. The next day, unhatched eggs and eggshells were removed from the dish to leave only newly hatched juveniles, which were defined as 0 days old. These juveniles were reared as described above until the appropriate age.

## Preparation of Cas9 complex (RNPs)

S.p. Cas9 Nuclease V3, CRISPR-Cas9 tracrRNA, and crRNAs were purchased from IDT. For each target genome region, the list of possible crRNAs and off-target information in *R. varieornatus* were retrieved using CRISPR direct (https://crispr.dbcls.jp/) and on-target scores of crRNAs were also obtained from the manufacturer's website (https://sg.idtdna.com/site/order/designtool/index/CRISPR_CUSTOM). Based on this information, appropriate crRNAs were selected as described in S6 Table. Cas9 complex (RNPs) was prepared as essentially described previously while increasing the concentration of each component [26]. Briefly, the mixture of crRNAs and tracrRNA (final 100 μM for each) was heated at 95°C for 5 min and then gradually cooled to room temperature. In a 10 μL scale, 1 μL of PBS (137 mM NaCl, 2.7 mM KCl, 10 mM $Na_2HPO_4$, 1.8 mM $KH_2PO_4$), 3 μL of 10 μg/μL Cas9 protein (IDT, final 3.0 μg/μL), and 6 μL of 100 μM gRNA (final 60 μM) were mixed and incubated at room temperature for 15–20 min to assemble Cas9 RNPs. Cas9 RNP solution was kept at -80°C until use.

## Injections

The injections were performed as described previously [26]. Briefly, the staged animals were anesthetized in 25 mM levamisole (Sigma) and mounted on an injection slide. The injection slide was placed on an inverted differential interference contrast microscope (Axiovert 405 M, Zeiss). A glass capillary needle was prepared from a glass capillary (GD-1, Narishige) using a needle puller (PC-10, Narishige). Cas9 RNP solution was filled into the glass capillary needle and injected into a body cavity of a tardigrade using FemtoJet 4i (Eppendorf) with the following settings: pi = 1500 hPa, ti = 0.20 s, and pc = 50 hPa. Successful injection was confirmed by swelling of the specimen. Injected individuals were recovered onto agar plates with sterilized water and maintained with food until they laid eggs ($G_0$). We collected $G_0$ eggs laid within 10 days after injection. Successfully hatched $G_0$ progeny were subjected to genotyping before or after laying eggs ($G_1$).

## Genotyping

Genomic PCR was performed as described in a previous study with some modifications [26]. Briefly, genomic DNA was extracted from each $G_0$ individual in 10 μL of protease K solution (500 μg/mL, in 0.5× KOD FX Neo PCR buffer; TOYOBO) by incubating at 60°C for 120 min. Protease K was then inactivated by heating at 95°C for 15 min. Genome samples were stored at -20°C until the PCR reaction. Primers were designed to include all target sites of crRNAs (S6 Table). The target regions were amplified from genome samples of approximately 0.5 to 2 μL using KOD FX Neo (TOYOBO) in a 10 μL reaction. PCR amplicons were examined by agarose gel electrophoresis (AGE). Genomic PCR products that were confirmed to be represented by a single band in AGE were subjected to direct Sanger sequencing after decomposing the remaining nucleotides and primers by treatment with 2U Exonuclease I (NEB) and 0.1U Shrimp Alkaline Phosphatase (NEB).

## Knock-in experiments

ssODNs were designed to introduce 11 separate single-nucleotide substitutions, two of which involved mutation of the PAM sequence (S6 Table). The designed ssODNs (189 bases) were synthesized by and purchased from IDT. RNP solution was prepared as described above and 10 μL of it was mixed with 1 μL of 14.5 μg/μL ssODNs (final 1.3 μg/μL) and used for injection. The solution was kept at -80°C until use.

## Phylogenetic analysis

Phylogenetic analysis of the ABCG family was performed essentially in accordance with a previously reported procedure [44]. Full-length protein sequences of 66 ABCG proteins of four insects and one tardigrade were retrieved from the NCBI protein database (S1 Data). As an outgroup, three ABCH protein sequences of *D. melanogaster* were included in the analysis. Those sequences were aligned using Muscle 3.8.425 [45]. A maximum likelihood analysis was performed with the substitution model LG using PhyML 3.3.20180621 as a plugin in Geneious software (Dotmatics) [46].

## Supporting information

**S1 Fig. Phylogenetic analysis of ABC transporter subfamily G.** Phylogenetic analysis using 66 ABCG proteins from a tardigrade (Rv, *Ramazzottius varieornatus*) and four insects (AG, *Anopheles gambiae*; Am, *Apis mellifera*; Dm, *Drosophila melanogaster*; Tc, *Tribolium castaneum*). Maximum likelihood analysis was performed with the substitution model LG. Red indicates the gene targeted by CRISPR-Cas9 in this study. Three proteins of ABC transporter subfamily H were used as an outgroup.
(TIF)

**S2 Fig. Confirmation of homozygosity of *RvY_01244* (*ABCG*) mutations in m2 and m3 mutants using the second set of PCR primers.** (A) Schematic representation of the second primer set (magenta arrows) with the structure of the *RvY_01244* (*ABCG*) gene, three crRNAs (brown arrows), and the original set of PCR primers (blue arrows). Green boxes represent exons and gray lines represent introns or intergenic regions. (B, C) Electropherograms in direct Sanger sequencing of genomic PCR amplicons amplified with the second primer set. (B) m2 carrying a 1-nt insertion. (C) m3 carrying a 3-nt deletion. No mixed peaks were detected, suggesting that these edits were present homozygously.
(TIF)

**S3 Fig. Comparison of appearance of *RvY_01244* (*ABCG*) mutant and $G_0$ individual carrying no edits.** Photographs of $G_0$ individuals carrying the m4 mutation (A) and no edits (B) at 16 days old. The m4 mutant exhibited a brown color and was apparently indistinguishable from the $G_0$ individual carrying no edits. Compared with wild-type individuals (Fig 1B and 1C), both of them were much smaller and had a relatively faint body color.
(TIF)

**S4 Fig. Electropherograms of direct Sanger sequencing in $G_0$ individuals with *RvY_13060* (*tps-tpp*) gene editing.** Electropherogram data corresponding to Fig 3C (A, 1-nt ins; B, 8-nt del; C, 483-nt del; D, 478-nt del) and Fig 3E (E, 8-nt + 3-nt del; F, 484-nt del). No mixed peaks were detected, suggesting that all of these $G_0$ individuals were homozygous mutants.
(TIF)

**S5 Fig. Confirmation of homozygosity of *tps-tpp* $G_0$ mutants using the second set of primers.** (A) Schematic representation of the second primer set (magenta arrows) with the structure of the *RvY_13060* (*tps-tpp*) gene, two crRNAs (brown arrows), and the original set of PCR primers (blue arrows). Green boxes represent exons and gray lines represent introns. (B, C) Electropherograms in direct Sanger sequencing of the genomic PCR amplicons amplified with the second primer set. Data correspond to Fig 3E (B, 1-nt ins; C, 8-nt + 3-nt del). No mixed peaks were detected, suggesting that these edits were present homozygously.
(TIF)

**S6 Fig. Heritability of the knock-in sequence to $G_2$ progeny.** Representative electropherograms of a $G_1$ individual (I-1) and three $G_2$ individuals derived from a perfectly knocked-in $G_0$ individual (I; S5 Table). REF represents the sequence of the unmodified genome, and the designed substitutions in ssODNs are shown as a KI design. The substituted bases are shown highlighted. All progeny exhibited clear single sequences without mixed peaks, indicating the heritability of the knock-in sequence as a homozygous mutation.
(TIF)

**S7 Fig. Comparison of appearance of *RvY_01244* (*ABCG*) knock-in mutant and $G_0$ individual carrying no edits.** Representative photographs of a knock-in individual carrying all of the 11 substitutions in the *RvY_01244 (ABCG)* gene (perfect substitutions; A) and an individual carrying no edits (B). The body color of knock-in individuals appeared comparable to that of those with no edits.
(TIF)

**S8 Fig. Proposed model for cytological process accounting for dominant generation of homozygous mutants in parthenogenetic tardigrades.**
(TIF)

**S1 Table. Concentrations of Cas9 and glycerol used in the related studies.**
(TIF)

**S2 Table. Effects of glycerol concentration on the survival of the injected tardigrades.**
(TIF)

**S3 Table. Hatchability of $G_1$ eggs laid by $G_0$ individuals carrying edited *tps-tpp*.**
(TIF)

**S4 Table. Hatchability of $G_1$ eggs laid by $G_0$ individuals without editing in *tps-tpp*.**
(TIF)

**S5 Table. Summary of heritability of knock-in alleles down to $G_2$ progeny.**
(TIF)

**S6 Table. Sequences of PCR primers (A), crRNAs (B), and ssODNs (C).**
(TIF)

**S1 Data. Accession numbers for ABCG and ABCH protein sequences used in the phylogenetic analysis.**
(XLSX)

## Acknowledgments

We thank Edanz (https://jp.edanz.com/ac) for English language editing.

## Author Contributions

**Conceptualization:** Koyuki Kondo, Takekazu Kunieda.

**Data curation:** Koyuki Kondo, Takekazu Kunieda.

**Formal analysis:** Koyuki Kondo.

**Funding acquisition:** Akihiro Tanaka, Takekazu Kunieda.

**Investigation:** Koyuki Kondo.

**Methodology:** Koyuki Kondo, Akihiro Tanaka.

**Project administration:** Takekazu Kunieda.

**Resources:** Akihiro Tanaka.

**Supervision:** Takekazu Kunieda.

**Validation:** Koyuki Kondo, Takekazu Kunieda.

**Visualization:** Takekazu Kunieda.

**Writing – original draft:** Koyuki Kondo, Takekazu Kunieda.

**Writing – review & editing:** Koyuki Kondo, Akihiro Tanaka, Takekazu Kunieda.

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
