## [Decision Letter · Decision Letter 0]

16 Feb 2024

Dear Dr Kunieda,

Thank you very much for submitting your Research Article entitled 'Single-step generation of homozygous knock-out/knock-in individuals in an extremotolerant parthenogenetic tardigrade using DIPA-CRISPR' to PLOS Genetics.

The manuscript was fully evaluated at the editorial level and by independent peer reviewers. The reviewers appreciated the attention to an important problem, but raised some substantial concerns about the current manuscript. Based on the reviews, we will not be able to accept this version of the manuscript, but we would be willing to review a much-revised version. We cannot, of course, promise publication at that time.

If you decide to revise the manuscript for further consideration at PLOS Genetics, please aim to resubmit within the next 60 days, unless it will take extra time to address the concerns of the reviewers, in which case we would appreciate an expected resubmission date by email to plosgenetics@plos.org.

We are sorry that we cannot be more positive about your manuscript at this stage. Please do not hesitate to contact us if you have any concerns or questions.

Yours sincerely,

Takaaki Daimon

Academic Editor

PLOS Genetics

Quanjiang Ji

Section Editor

PLOS Genetics

Comments from the Associate Editor:

As you will see below, the comments of the three reviewers are split: #1 and #3 are basically satisfied with the novelty and significance of this study, but #2 is not. I would like to suggest that the authors revise the manuscript according to the reviewers’ comments, with a stronger emphasis on the importance of tardigrade biology and the significance of this work within it.

1. As pointed out by the reviewers, the language of this manuscript should be extensively improved.

2. As pointed out by reviewer #3, a weakness of this paper would be the lack of in-depth functional studies of the target genes. However, since this paper is submitted to the Methods section/category, additional functional studies on a different/new target gene would not be necessary. Instead, I would suggest that the authors discuss more about the results of the gene editing experiments. For example, it is not clear whether the authors found some external phenotypes in the RvY_01244 KO animals. It is also unclear whether this gene is an ortholog of insect white genes. As insects have multiple white homologs (e.g., scarlet and brown in Drosophila, which form a heterodimer with white), and some species have multiple copies of white paralogs, the authors should carefully validate the characteristics of RvY_01244 and whether it is a tardigrade white gene.

3. For readers not familiar with tardigrades, it would be nice if the authors could show the pictures of tardigrades being injected or oocytes/eggs being developed or laid.

Reviewer's Responses to Questions

**Comments to the Authors:**

Reviewer #1: In this manuscript, Kondo et al demonstrate germline CRISPR (i.e., heritable edits) for the first time in the tardigrade phylum. The Kunieda lab had previously developed CRISPR in somatic cells, and here they have tried to extend this to heritable, germline edits by injecting Cas9 RNPs into the body cavity of tardigrades during late oogenesis, as has worked in insects. They first injected glycerol at various concentrations as a precursor to injecting a commercial Cas9 in glycerol, and they showed that about half of injected animals survive a concentration of glycerol that enables a similar concentration of Cas9 to be injected as was used in insects. Then they injected Cas9, tracRNA and crRNA as an RNP complex, and showed that injecting 8-10 days after hatching resulted in gene edits. This worked targeting a second gene as well, and they showed multiple kinds of edits, some homozygous edits, and inheritance of edits 1-2 generations later. Lastly, they showed that co-injecting single-stranded oligo DNA could be used to knock in at least short sequences.

The development of germline CRISPR (i.e., heritable edits) is a big step forward for an emerging model organism, and this had never been done before for any tardigrade (an entire phylum of animals). And there's broad interest in the use of tardigrades – arguably the animal kingdom's most extreme survivor – for understanding how biological materials can survive extremes. The experiments appear well thought out and reported clearly. I'm thrilled to see the development reported here, and I have only minor concerns.

Minor concerns:

1. The number of animals genotyped suggests that PCR often failed, which gives me just a little concern about whether PCR failures could have resulted in apparent homozygosity in what were actually heterozygotes. This concern could be taken care of by re-PCRing from some of the samples using a few pairs of primers (instead of just one primer pair) and seeing if the results are the same (and moving primers further out, to amplify longer sequences, would reveal whether edits can be found a little further out from the targeted sites).

2. The manuscript needs English editing, and I would argue would benefit tremendously from a good scientific copy editor. The ideas expressed are interesting, and I found the Discussion especially thoughtful and articulate, but the mechanical issues in the writing will be an obstacle for many readers.

3. Hypsibius exemplaris is described as having relatively weak tolerance ability. Hypsibius exemplaris requires slow drying to survive desiccation, and this often fails, but once in tun state, I'm not aware of any studies showing that they are significantly less resistant than other tardigrades to extremes. And Hypsibius exemplaris survives irradiation as well as other resistant tardigrades do. I think the authors mean to say that Hypsibius does not survive rapid desiccation, and not that their tolerance abilities are weak.

4. p. 13, line 275: The supplementary figure cited does not show the relevant data. Also, statements are made at the end of this paragraph without data shown.

5. p. 14, line 297: This paper speculated that storage cells synthesize yolk, but to my knowledge this has never been demonstrated (and there is not yet a marker for yolk protein, to my knowledge).

6. p. 15, line 328: Comparing knock-in and knock-out efficiency: Has a statistical test been done to see if this is a real difference? I suggest doing an appropriate test and then describing the two efficiencies as either different or indistinguishable.

Reviewer #2: Kondo et al. provide a first report on gene-edited GO progeny of tardigrades using the DIPA-CRISPR technique. They created knockouts of the white and tps-tpp gene, and also created a knock-in for the white-gene using ssODNs. Materials and methods are clear, and Discussion is to the point. I did not find any major flaws in the manuscript (see some comments below). However, as the CRISPR technique that was used to genetically transform tarfigrades is largely based on a previously published method (Shirai et al. 2022, Cell Reports), the research in the manuscript can, in my opinion, not be considered as a major advancement in the field and, hence, might better fit in another journal than PLoS Genetics.

Comments:

line 59: In a previous study,...

line 124: it is not clear to me whether RvY_01244 is an 1:1 ortholog of Drosophila white, or does the tardigrade has multiple white paralogs? Authors should include a ABCG phylogeny to demonstrate orthology

line 129-135: it is not clear to me what the phenotype is of w-m1, w-m2 and w-m3, this should be mentioned in the results section (including photographs of the w mutants)

line 176: as with the white gene; authors should construct a phylogenetic tree, displaying orthology between tardigrade tps-tpp and tps-tpp of other species; In addition, is the tps-tpp gene of R. varieorantus horizontally transferred? If so, make this more clear in the results section.

line 316: ... undergoes a..

line 319/320: mutants of haplodiploid insects/arthropods can also be obtained in a single step, see e.g. 10.1016/j.ibmb.2023.104068 (spider mites and thrips) or 10.1089/crispr.2019.0067 (whiteflies); authors should integrate these studies in the Discussion section

Reviewer #3: The current study by Kondo et al explores the use of gene editing techniques, specifically CRISPR-Cas9 and DIPA-CRISPR, in tardigrades, microscopic organisms known for their extreme resilience. The authors report the successful application of gene editing methods to modify genes in tardigrades, in particular with focus on the anhydrobiotic and extremotolerant species Ramazzottius varieornatus. They adapted the Direct Parental CRISPR (DIPA-CRISPR) method, initially developed for insects, to inject Cas9 ribonucleoproteins (RNPs) into female tardigrades. This resulted in gene-edited progeny, demonstrating the method's effectiveness. The study also explored gene knock-in trials with single-stranded oligodeoxynucleotides (ssODNs), achieving notable efficiency. This research provides significant insights into the molecular mechanisms of tardigrades' resilience and opens new avenues for functional genomics research in these organisms.

Overall, I find the subject of the manuscript to be interesting and relevant. The paper is for the most part well written; however, the manuscript would benefit from more careful copy editing, as there are many instances of wrong word-/verb-use making it difficult to understand what the authors mean. My perhaps biggest critique is that this is largely a descriptive methods paper, rather than a detailed functional study, which is otherwise expected by a PLoSGen paper. However, given the novelty and difficulty in performing genome editing in these animals, this is perhaps to be expected. Nevertheless, I would encourage the authors to consider adding more functional data illustrating the utility of this approach (see below), as the whole methodology argues that this would be entirely feasible.

Major comments:

1. Given that the authors report that this tardigrade species reproduce parthenogenetically, I was wondering if they allowed the w-m1-3 mutants to lay eggs prior to harvesting their DNA for sequencing (if there are no WT alleles present I assume the mutation was introduced in the germline, as one would otherwise observe a mosaic pattern)? Are these mutant lines still available, and if so, did the authors report any phenotypes associated with gene mutation, i.e. pigment defects or similar? I see later on that the authors did in fact attempt to propagate the F1 generation following their knock-in protocol, suggesting that this is a viable approach. However, again, there are no attempts made of performing phenotypic characterization.

If the authors could show, either 1) by knock-out of a non-essential gene followed by in-depth phenotypic characterization, or alternatively 2) by knock-in of e.g. GFP under control by a promotor of choice, it would significantly elevate the utility and impact of this paper.

Minor comments:

L32 Do the authors mean “conducive” instead of “conductive”?

L59-63 Consider revising this sentence as it is long and difficult to follow.

L68 “flour” instead of “floor”

L124 I assume the authors mean that RvY_01244 is predicted to encode a homologue of Drome-white? What is the degree of sequence homology between these two genes? To what extent are the core functional motifs of this ABCG2 transporter conserved, i.e. what is the data supporting that these genes are homologous? It could be interesting to show how this famous gene has changed over the course of evolution.

L130 Add “for” after “amplicons”.

Figures.

Figure 1 You mention in the result section that you examine injection into animals that are 5-10 days old, yet the figure shows 7-10 days. Which one is it? Please correct text and figure accordingly.

**Have all data underlying the figures and results presented in the manuscript been provided?**

Reviewer #1: Yes

Reviewer #2: **No: **see comment on w-m1, w-m2 and w-m3 mutants

Reviewer #3: None

PLOS authors have the option to publish the peer review history of their article (what does this mean?). If published, this will include your full peer review and any attached files.

If you choose “no”, your identity will remain anonymou

---

## [Decision Letter · Decision Letter 1]

8 May 2024

Dear Dr Kunieda,

Thank you very much for submitting your Research Article entitled 'Single-step generation of homozygous knockout/knock-in individuals in an extremotolerant parthenogenetic tardigrade using DIPA-CRISPR' to PLOS Genetics.

The manuscript was fully evaluated at the editorial level and by independent peer reviewers. The reviewers appreciated the attention to an important topic but identified some concerns that we ask you address in a revised manuscript.

We therefore ask you to modify the manuscript according to the review recommendations. Your revisions should address the specific points made by each reviewer.

Yours sincerely,

Takaaki Daimon

Academic Editor

PLOS Genetics

Quanjiang Ji

Section Editor

PLOS Genetics

The submitted revised manuscript has been evaluated by the three original reviewers.

The authors have done a good job in revising the manuscript, and as you will see below, two reviewers are very positive and satisfied with the revision, although reviewer #2 raised some minor concerns.

Therefore, I'd like to ask the authors to revise the manuscript.

Reviewer's Responses to Questions

**Comments to the Authors:**

Reviewer #1: I am content with all of the edits that the authors made in response to my concerns. The development of heritable CRISPR-based genome editing is a big step forward for a phylum where it had never been done before and will open up studies that are not possible in other organisms, given tardigrades' unique ability to survive some remarkable extremes.

Reviewer #2: The authors addressed most of my comments (e.g. ABCG orthology) in an adequate way, and made more clear that their study addressed a longstanding challenge in the field (i.e. heritable gene editing of tardigrades). However, I am not completely satisfied with all replies. Please find a below some remarks to their responses:

1. In their reply to my first comment the authors state that DIPA-CRISPR has only been tested in insects. This seems not correct as in Dermauw et al. 2020 adult females of the spider mite T. urticae (Arthropoda:Chelicerata) were injected with Cas9/sgRNAs and KO mutants were succesfully generated. It was only in 2022, that this technique was termed DIPA-CRISPR by Shirai et al. 2022. Authors should correct their manuscript accordingly.

2. description of "mutant phenotypes": what is the difference between growth arrest (m1, m2) and growth retardation (m4)? Authors should be more clear.

3.L379/L380: "In these cases, multiple crossings and selections are needed to establish homozygous mutant strains". This statement is not correct as in a recent study (10.1016/j.ibmb.2023.104068) it was shown that homzygous mutants could be generated via Cas9/sgRNA injection of fertilized females. Authors should rephrase.

Reviewer #3: The authors have addressed all concerns raised successfully and I have no further comments.

**Have all data underlying the figures and results presented in the manuscript been provided?**

Reviewer #1: Yes

Reviewer #2: Yes

Reviewer #3: Yes

PLOS authors have the option to publish the peer review history of their article (what does this mean?). If published, this will include your full peer review and any attached files.

Reviewer #1: No

Reviewer #2: No

Reviewer #3: **Yes: **Kenneth Veland Halberg

---

## [Editor Report · Decision Letter 2]

10 May 2024

Dear Dr Kunieda,

We are pleased to inform you that your manuscript entitled "Single-step generation of homozygous knockout/knock-in individuals in an extremotolerant parthenogenetic tardigrade using DIPA-CRISPR" has been editorially accepted for publication in PLOS Genetics. Congratulations!

Yours sincerely,

Takaaki Daimon

Academic Editor

PLOS Genetics

Quanjiang Ji

Section Editor

PLOS Genetics

Comments from the reviewers (if applicable):

The authors have done a good job revising the manuscript.

It is now ready for publication.

**Data Deposition**

http://datadryad.org/submit?journalID=pgenetics&manu=PGENETICS-D-24-00026R2

**Press Queries**

---

## [Editor Report · Acceptance letter]

21 May 2024

PGENETICS-D-24-00026R2 

Single-step generation of homozygous knockout/knock-in individuals in an extremotolerant parthenogenetic tardigrade using DIPA-CRISPR 

Dear Dr Kunieda, 

We are pleased to inform you that your manuscript entitled "Single-step generation of homozygous knockout/knock-in individuals in an extremotolerant parthenogenetic tardigrade using DIPA-CRISPR" has been formally accepted for publication in PLOS Genetics! Your manuscript is now with our production department and you will be notified of the publication date in due course.

With kind regards,

Anita Estes

PLOS Genetics

On behalf of:
